# Epidemiology of vulvovaginal candidiasis in China: a six-year retrospective single-center study from a tertiary obstetrics and gynecology hospital in Shanghai

Yisheng Chen,[1] Loukaiyi Lu,[1] Qiang Wang,[1] Xufan Luo,[1] Jing Gao,[1] Chunmei Ying[1]

**ABSTRACT**   Vulvovaginal candidiasis (VVC) represents a global health issue. However, epidemiological data on VVC in eastern China are scarce, limited, and outdated. To address this gap, we conducted a six-year retrospective study (2019–2024), investigating the epidemiological characteristics, species distribution, and antifungal susceptibility patterns of VVC patients in Shanghai, China. VVC was confirmed based on clinical symptoms combined with laboratory examination. *Candida* species were identified by MALDI-TOF MS, and *in vitro* antifungal susceptibility testing (AFST) was carried out using the ATB FUNGUS 3 method. The epidemiological changes over time were analyzed using the $\chi 2$ test for trend. In total, 7,894 VVC episodes were identified among 7,167 individuals. Of these, 39 patients (0.54%) exhibited mixed *Candida* infections. *Candida albicans* was the predominant species (77.26%), followed by *Candida glabrata* (17.63%), *Candida parapsilosis* (1.53%), *Candida tropicalis* (1.17%), and *Candida krusei* (1.00%). Other species were rare, each accounting for less than 1.0%. VVC shows high prevalence in our region, with *C. albicans* as the most prevalent pathogen, followed by *C. glabrata*. And an increasing trend in *C. glabrata* VVC was observed, accompanied by consistently high rates of voriconazole resistance. Fluconazole resistance was observed in 18.82% of *C. albicans* with cross-resistance to other azoles. Thus, our findings highlight the significance of *Candida* species identification and accurate diagnosis in VVC, underscoring the necessity for targeted antifungal management.

**IMPORTANCE** Vulvovaginal candidiasis is a widespread condition among women, associated with significant emotional, physical, and economic burdens, and presents ongoing challenges in diagnosis and management. Our six-year Shanghai study reveals *Candida* species caused 9.66% of vaginitis cases, with *C. albicans* remaining predominant and showing alarming fluconazole resistance (18.82%). Particularly concerning is the rising trend of non-albicans *Candida*, especially *C. glabrata*. These findings critically emphasize the necessity of implementing routine species identification and continuous antifungal susceptibility testing in China to guide targeted therapy, curb resistance spread, and optimize VVC management strategies.

**KEYWORDS** vulvovaginal candidiasis, epidemiology, *Candida albicans*, *Candida glabrata*, antifungal resistance

Vulvovaginal candidiasis (VVC) is the second most common vaginal infection worldwide, affecting 75% of women at least once in their lifetime, with 5%–9% developing recurrent vulvovaginal candidiasis (RVVC) (1). Predisposing factors for VVC include diabetes mellitus, contraceptive use, broad-spectrum antibiotic therapy, pregnancy, and host genetics background (2). Beyond physical discomfort, VVC and RVVC significantly reduce quality of life for women, leading to psychosocial issues such

**Peer Reviewer** Eman Adel Elansoury, Mansoura University, Mansoura, Dakahlia, Egypt

Address correspondence to Chunmei Ying, ycmzh2012@163.com, or Jing Gao, gaojing1511@fckyy.org.cn.

Yisheng Chen and Loukaiyi Lu contributed equally to this article. Author order was determined on the basis of contributions.

The authors declare no conflict of interest.

See the funding table on p. 9.

as increased stress, anxiety, depression, and social isolation (3). Despite its considerable clinical and psychosocial impact, the epidemiological profile of VVC is still poorly characterized across many parts of the world, particularly throughout the majority of the Asia-Pacific region (4).

Although *Candida albicans* remains the predominant pathogen responsible for VVC accounting for 85%–95% of cases (5), non-albicans *Candida* (NAC) species such as *C. glabrata*, *C. tropicalis*, and *C. krusei* are increasingly prevalent worldwide and now constitute 10%–45% of infections (6–8). This epidemiological shift varies geographically. Antifungal agents commonly used in the treatment of VVC include azoles, polyenes, and flucytosine. However, studies have reported the increasing fluconazole resistance rates of VVC patients in multiple geographical regions (9–12). Self-medication and empirical treatment may exacerbate the development of antifungal resistance (13, 14). Although amphotericin B retains efficacy, its use is limited by significant kidney toxicity and high cost (15). Similarly, flucytosine is also expensive in many developing countries, and monotherapy with flucytosine carries a significant risk of rapidly emerging resistance (16). Treating drug-resistant vaginitis remains a clinical challenge. Given the critical dependence of empirical treatment on local epidemiological data, it is imperative to regularly monitor regional variations of VVC isolates (17).

We conducted a six-year retrospective study analyzing 7,894 clinical isolates of patients with VVC in Shanghai, aiming to provide updated epidemiological data and antimicrobial susceptibility profiles to guide evidence-based VVC management in our region.

## RESULTS

### Species distribution of VVC

In total, 74,206 symptomatic women were enrolled during the six-year study period. Among these, 7,894 *Candida* strains were isolated from 7,167 patients, yielding a prevalence of 9.66%. Moreover, 39 patients (39/7,167, 0.54%) exhibited mixed *Candida* infections. A total of 21 distinct *Candida* species were identified by MALDI-TOF (Fig. 1A). *C. albicans* was the most common species (6,099/7,894, 77.26%), followed by *C. glabrata* (1,392/7,894, 17.63%), *C. parapsilosis* (121/7,894, 1.53%), *C. tropicalis* (92/7,894, 1.17%), *C. krusei* (79/7,894, 1.00%), *C. lusitaniae* (11/7,894, 0.14%), and *C. metapsilosis* (9/7,894, 0.11%). Other species were rare, each representing less than 0.1%, including *C. nivariensis* (three strains), *C. dubliniensis* (two strains), *C. pelliculosa* (two strains), *C. bracarensis* (one strain), *Hansenula* (one strain), *C. guilliermondii* (one strain), *Pichia manshurica* (one strain), *C. orthopsilosis* (one strain), *Pichia norvegensis* (one strain), *C. norvegensis* (one strain), *C. kefyr* (one strain), *C. famata* (one strain), *C. sphaerica* (one strain), and *Rhodotorula mucilaginosa* (one strain). Additionally, this study identified 73 strains of *S. cerevisiae*, accounting for 0.92%. In the cohort of 39 patients with mixed *Candida* infections, *C. albicans* remained the predominant species (29/78, 37.18%). Specifically, the most frequent co-infection pattern was *C. albicans* and *C. glabrata* (18 cases, 46.15%), followed by *C. albicans* and *C. tropicalis* (6 cases, 15.38%), *C. glabrata* and *S. cerevisiae* (5 cases, 12.82%), *C. albicans* and *C. parapsilosis* (4 cases, 10.26%), and *C. albicans* and *C. krusei* (2 cases, 5.13%). Additionally, several rare co-infection patterns were observed (one case each, 2.56%), including *C. glabrata* and *R. mucilaginosa*, *S. cerevisiae* and *C. sphaerica*, *C. glabrata* and *C. parapsilosis*, and *C. lusitaniae* and *C. parapsilosis*.

We also examined the annual prevalence of various *Candida* spp. throughout the study period (Fig. 1B). The results showed that the prevalence of *C. albicans* exhibited an inverse trend, declining from 83.29% to 70.99% ($\chi2 = 102.15$, $P < 0.01$) while NAC prevalence exhibited an increasing trend. We also observed a significant increase from 12.00% to 23.38% in *C. glabrata* VVC ($\chi2 = 96.89$, $P < 0.01$).

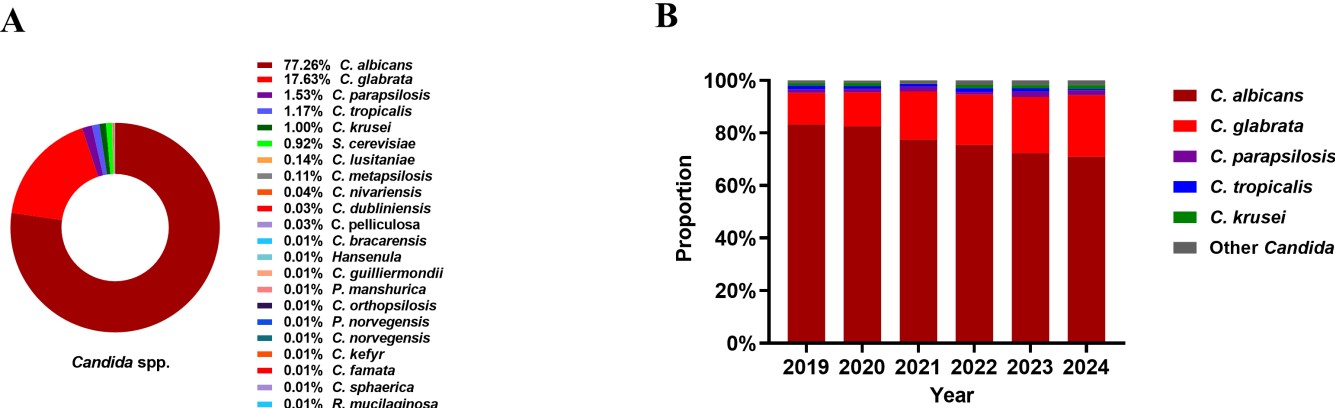

**FIG 1** Distribution of *Candida* proportions. (A) Prevalence of *Candida* in the overall patient cohort during the study. (B) Annual distribution of *Candida* spp. infections (2019–2024). "Other *Candida*" included *S. cerevisiae* (*n* = 73), *C. lusitaniae* (*n* = 11), *C. metapsilosis* (*n* = 9), *C. nivariensis* (*n* = 3), *C. dubliniensis* (*n* = 2), *C. pelliculosa* (*n* = 2), *C. bracarensis* (*n* = 1), *Hansenula* (*n* = 1), *C. guilliermondii* (*n* = 1), *Pichia manshurica* (*n* = 1), *C. orthopsilosis* (*n* = 1), *Pichia norvegensis* (*n* = 1), *C. norvegensis* (*n* = 1), *C. kefyr* (*n* = 1), *C. famata* (*n* = 1), *C. sphaerica* (*n* = 1), and *R. mucilaginosa* (*n* = 1). It was consistent with the usage in Fig. 2.

## Age distribution of VVC patients

The median age of population with VVC was 32 years (IQR [28–38]), with ages ranging from 5 to 82 years. The age distribution was considerably skewed toward women of reproductive age. Although the most prevalent *Candida* species was *C. albicans* across all age groups, its distribution was different. Specifically, in the group of patients aged ≤18 years, the prevalence of *C. albicans* was 83.02%, while 80.01%, 74.39%, and 64.56% for those aged 19–35, 36–45, and ≥46 years old, respectively ($\chi2 = 91.74$, $P < 0.01$). In contrast, NAC species, particularly *C. glabrata*, showed an increased trend with prevalence of 7.55%, 15.67%, 21.06%, and 23.09% for those aged ≤18, 19–35, 36–45, and ≥46 years old, respectively ($\chi2 = 44.95$, $P < 0.01$) (Table 1).

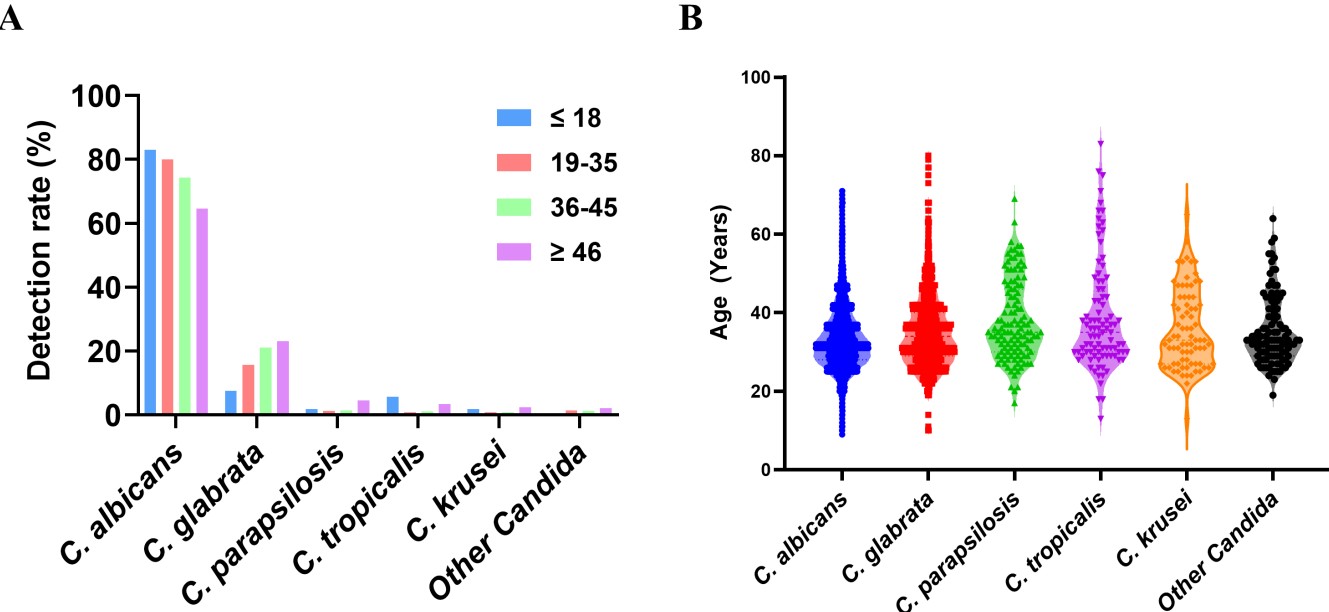

**FIG 2** Age distribution of VVC patients during the study period. (A) Isolation rates of *Candida* spp. from reproductive tract samples of women, separated by their age groups. (B) Age distribution among patients diagnosed with VVC. The width of the violin plot represented the patient count per age group.

**TABLE 1** *Candida* spp. among VVC patients in separated age groups (*n*, %])

| Age | *C. albicans* | *C. glabrata* | *C. parapsilosis* | *C. tropicalis* | *C. krusei* | Other *Candida* |
|---|---|---|---|---|---|---|
| ≤18 | 44 (83.02) | 4 (7.55) | 1 (1.89) | 3 (5.66) | 1 (1.89) | 0 (0) |
| 19–35 | 4,111 (80.01) | 805 (15.67) | 62 (1.21) | 44 (0.86) | 44 (0.86) | 72 (1.40) |
| 36–45 | 1,505 (74.39) | 426 (21.06) | 27 (1.33) | 22 (1.09) | 18 (0.89) | 25 (1.24) |
| ≥46 | 439 (64.56) | 157 (23.09) | 31 (4.56) | 23 (3.38) | 16 (2.35) | 14 (2.06) |

## Antifungal susceptibility testing

The antifungal susceptibility testing (FUNGUS 3) was conducted on all 7,894 isolates obtained from women with VVC. As summarized in Table 2, all *Candida* isolates demonstrated excellent susceptibility to amphotericin B, exhibiting low MIC values (geometric mean MIC, GM MIC = 0.26 mg/L). For 5-fluorocytosine (5-FC), *C. krusei* isolates demonstrated reduced susceptibility, with elevated MIC values (GM MIC = 7.16 mg/L). While, all other *Candida* species maintained lower MIC values for 5-FC. In contrast, marked variations were observed among azole antifungals against vaginal *Candida* isolates, with fluconazole demonstrating higher GM MIC values (15.85 mg/L) compared to itraconazole (0.83 mg/L) and voriconazole (1.16 mg/L). And the $MIC_{50}$/$MIC_{90}$ values differed by 3–4 twofold dilutions, respectively. Moreover, NAC species exhibited universally higher MIC values, with *C. tropicalis* displaying the highest resistance profiles (GM MIC: 43.90, 1.43, and 2.91 mg/L; $MIC_{90}$: >128, 8, and 16 mg/L for fluconazole, itraconazole, and voriconazole, respectively). While, *C. glabrata* showed GM MICs of 7.01, 0.90, and 0.48 mg/L for those azole drugs, respectively. In contrast, *C. parapsilosis* maintained relatively lower susceptibility thresholds to all three azole antifungals.

Further analysis classified the 7,894 vaginal *Candida* isolates into WT (comprising S and I/SDD strains) and NWT/R strains (Table 2). The majority of the tested strains maintained susceptibility to amphotericin B, whereas 210 isolates (2.66%) exhibited 5-FC resistance. The overall resistance/NWT rates of fluconazole, itraconazole, and voriconazole were 16.35%, 33.63%, and 15.86%, respectively. Notable variations in resistance patterns were observed among *Candida* species. For *C. albicans*, the NWT rates of fluconazole, itraconazole, and voriconazole were 18.82%, 41.84%, and 14.49%, respectively. Among NAC species, *C. glabrata* exhibited relatively low fluconazole (26/1,392, 1.87%) and itraconazole (74/1,392, 5.32%) but elevated voriconazole (338/1,392, 24.28%) NWT/resistance rates. *C. tropicalis* showed the highest resistance rates to all three azoles (33.70%, 22.83%, and 27.17%, respectively). Although *C. krusei* is intrinsically resistant to fluconazole, its NWT rates for itraconazole and voriconazole were identical, both accounting for 2.53%. And *C. parapsilosis* demonstrated minimal resistance to voriconazole and fluconazole (1.56% each).

Remarkably, one multidrug-resistant (MDR) *C. albicans* isolate (resistant to ≥3 antifungal classes) was detected among the tested *Candida* isolates. Furthermore, 63 isolates (0.80%) exhibited dual resistance to both azoles and 5-FC, with *C. albicans* being the predominant species (*n* = 46, 73.02%), followed by *C. krusei* (*n* = 9, 14.29%) and *C. glabrata* (*n* = 8, 12.70%). Additionally, 2,953 isolates (37.41%) were resistant/NWT to two of the three tested azole antifungals. This resistance patterns revealed significant cross-resistance among azole antifungals.

## Temporal trends in azole resistance of *C. albicans* and *C. glabrata*

We further analyzed the six-year dynamic changes in GM MIC values and resistance rates of the most prevalent vaginal *Candida* species (*C. albicans* and *C. glabrata*) to three azoles. For *C. albicans*, the resistance rate to fluconazole significantly decreased from 26.14% (361/1,381) in 2019 to 17.37% (173/996) in 2024 ($\chi^2$ = 25.56, *P* < 0.01), accompanied by a decline in GM MIC from 26.45 mg/L to 17.86 mg/L (Fig. 3A). Although the GM MIC of itraconazole against *C. albicans* decreased from 1.09 mg/L (2019) to 0.87 mg/L (2024), the NWT rate remained persistently high (Fig. 3B). Similarly, voriconazole resistance showed a significant reduction from 21.00% (290/1,381) in 2019 to 12.75%

**TABLE 2** Profile of *in vitro* susceptibility to five antifungals of *Candida* spp. from VVC patients in Shanghai[a]

| Species (no.) | Antifungal agent | MIC (range) (mg/L) | GM MIC (mg/L) | MIC$_{50}$/MIC$_{90}$ (mg/L) | WT (n [%]) | | NWT/resistance (n [%]) |
|---|---|---|---|---|---|---|---|
| | | | | | S | I/SDD | |
| *C. albicans* | 5-FC | <4 to >16 | 2.65 | <4/<4 | 5,889 (96.56) | 49 (0.8) | 161 (2.64) |
| (n = 6,099) | AMB | <0.5 to >16 | 0.26 | <0.5/<0.5 | 6,097 (99.97) | | 2 (0.03) |
| | ITC | <0.12 to >4 | 0.82 | 0.12/1 | 3,547 (58.16) | | 2,552 (41.84) |
| | VRC | <0.06 to >8 | 1.34 | 0.12/1 | 3,071 (50.35) | 2,144 (35.15) | 884 (14.49) |
| | FLC | <1 to >128 | 17.84 | 2/8 | 3,850 (63.13) | 1,101 (18.05) | 1,148 (18.82) |
| *C. glabrata* | 5-FC | <4 to >16 | 2.57 | <4/<4 | 1,345 (96.62) | 9 (0.65) | 38 (2.73) |
| (n = 1,392) | AMB | <0.5 to 1 | 0.25 | <0.5/<0.5 | 1,392 (100) | | 0 (0) |
| | ITC | <0.12 to >4 | 0.90 | 0.25/2 | 1,318 (94.68) | | 74 (5.32) |
| | VRC | <0.06 to >8 | 0.48 | 0.12/0.5 | 1,054 (75.72) | | 338 (24.28) |
| | FLC | <1 to >128 | 7.01 | 2/8 | | 1,366 (98.13) | 26 (1.87) |
| *C. parapsilosis* | 5-FC | <4 to 2 | 2 | <4/<4 | 121 (100) | | 0 (0) |
| (n = 121) | AMB | <0.5 to 1 | 0.26 | <0.5/<0.5 | 121 (100) | | 0 (0) |
| | ITC | <0.12 to 1 | 0.09 | <0.12/<0.12 | 120 (99.17) | | 1 (0.83) |
| | VRC | <0.06 to 1 | 0.06 | <0.06/0.06 | 117 (96.69) | 2 (1.65) | 2 (1.65) |
| | FLC | <1 to >128 | 2.96 | <1/2 | 117 (96.69) | 2 (1.65) | 2 (1.65) |
| *C. tropicalis* (n = 92) | 5-FC | <4 to 4 | 2.04 | <4/<4 | 90 (97.83) | 2 (2.17) | |
| | AMB | <0.5 to 0.5 | 0.26 | <0.5/<0.5 | 92 (100) | | 0 (0) |
| | ITC | <0.12 to >4 | 1.43 | 0.12/8 | 71 (77.17) | | 21 (22.83) |
| | VRC | <0.06 to >8 | 2.91 | 0.12/16 | 52 (56.52) | 15 (16.30) | 25 (27.17) |
| | FLC | <1 to >128 | 43.90 | 2/>128 | 56 (60.89) | 5 (5.43) | 31 (33.70) |
| *C. krusei* (n = 79) | 5-FC | <4 to >16 | 7.16 | 8/16 | 19 (24.05) | 51 (64.56) | 9 (11.39) |
| | AMB | <0.5 to 2 | 0.32 | <0.5/0.5 | 79 (100) | | 0 (0) |
| | ITC | <0.12 to 2 | 0.49 | 0.5/1 | 77 (97.47) | | 2 (2.53) |
| | VRC | <0.06 to 4 | 0.46 | 0.25/1 | 70 (88.61) | 7 (8.86) | 2 (2.53) |
| | FLC | 2 to >128 | 20.68 | 16/32 | | | IR |
| Other *Candida* spp. | 5-FC | <4 to >16 | 2.45 | <4/<4 | 108 (97.30) | 1 (0.90) | 2 (1.80) |
| (n = 111) | AMB | <0.5 to 0.5 | 0.25 | <0.5/<0.5 | 111 (100) | | 0 (0) |
| | ITC | <0.12 to >4 | 0.58 | 0.25/1 | 101 (91.00) | 5 (0.95) | 5 (0.95) |
| | VRC | <0.06 to 2 | 0.14 | 0.06/0.25 | 110 (99.10) | | 1 (0.90) |
| | FLC | <1 to >128 | 5.00 | 2/4 | 110 (99.10) | | 1 (0.90) |
| Total (n = 7,894) | 5-FC | <4 to >16 | 2.66 | 2/2 | 7,572 (95.92) | 112 (1.42) | 210 (2.66) |
| | AMB | <0.5 to >16 | 0.26 | <0.5/<0.5 | 7,892 (99.97) | | 2 (0.03) |
| | ITC | <0.12 to >4 | 0.83 | 0.12/1 | 5,234 (66.30) | 5 (0.06) | 2,655 (33.63) |
| | VRC | <0.06 to >8 | 1.16 | 0.12/1 | 4,474 (56.68) | 2,168 (27.46) | 1,252 (15.86) |
| | FLC | <1 to >128 | 15.85 | 2/8 | 4,129 (52.31) | 2,474 (31.34) | 1,291 (16.35) |

[a]5-FC, 5-fluorocytosine; AMB, amphotericin B; FLC, fluconazole; ITC, itraconazole; VRC, voriconazole; GM, geometric mean; MIC, minimum inhibitory concentration; S, susceptible; I, intermediate; SDD, susceptible dose-dependent; R, resistant; WT, wild type; NWT, non-wild type.

(127/996) in 2024 ($\chi^2$ = 27.22, *P* < 0.01), with concurrent GM MIC reduction from 1.87 mg/L to 1.43 mg/L (Fig. 3C). In contrast, although *C. glabrata* exhibited no significant changes in NWT rates to any azoles during the study period (Fig. 3D and E), voriconazole maintained consistently high NWT rates against *C. glabrata* (20.22%–27.50%) over various years (Fig. 3F).

## DISCUSSION

As a developing country with a large population, a substantial number of Chinese women are affected by VVC. Existing epidemiological literature on VVC remains limited with over 50% of studies conducted before 2010, including in our region (17). During this six-year retrospective period, the prevalence of VVC among symptomatic women in Shanghai (9.66%) is relatively low compared to rates reported in most other regions worldwide (12%–78%) (18, 19). Additionally, mixed *Candida* infections were only

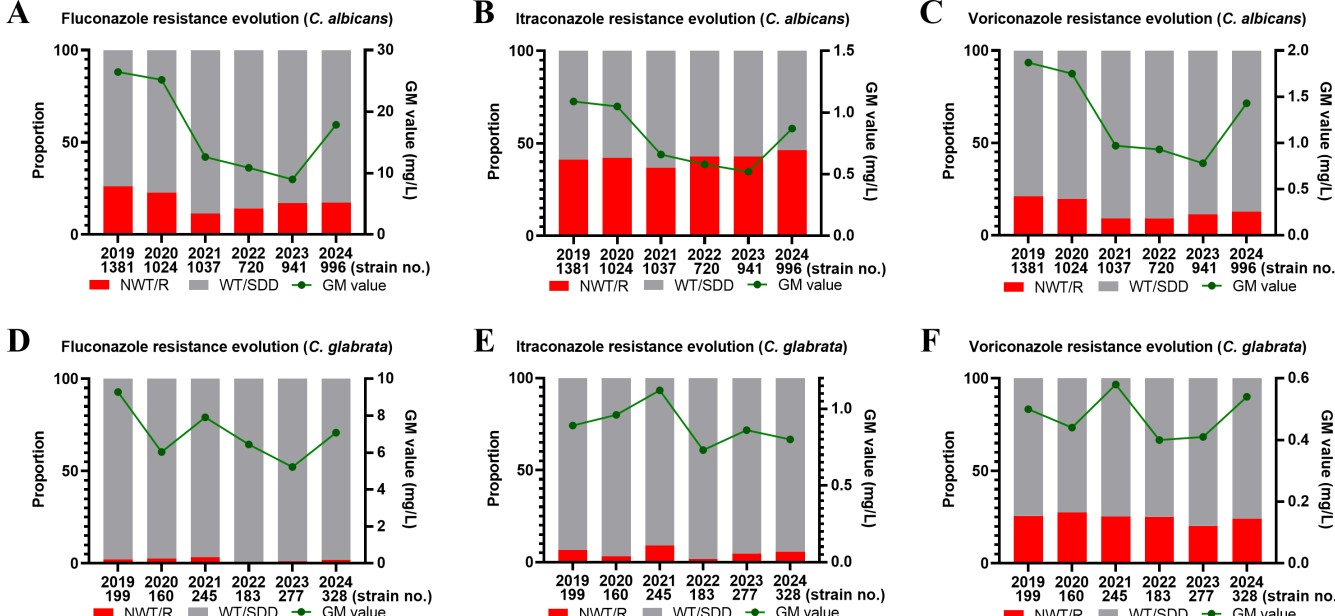

**FIG 3** Drug resistance evolution to three azoles in *C. albicans* and *C. glabrata* from VVC patients. (A) Fluconazole resistance rate of *C. albicans*. (B) Itraconazole NWT rate of *C. albicans*. (C) Voriconazole resistance rate of *C. albicans*. (D) Fluconazole resistance rate of *C. glabrata*. (E) Itraconazole NWT rate of *C. glabrata*. (F) Voriconazole NWT rate of *C. glabrata*. Abbreviations: FLC, fluconazole; ITC, itraconazole; VRC, voriconazole; GM value, geometric mean of MIC; SDD, susceptible dose-dependent; R, resistant; WT, wild type; NWT, non-wild type.

identified in 39 patients (0.54%), which is also much lower than previous findings (8.5%–14.1%) (18, 20). This variation may be attributed to differences in geographical location, study populations, and diagnostic methodologies (21). Among 7,894 isolates, *C. albicans* accounted for 77.26%, followed by *C. glabrata*, detected in 17.63% of cases, which are consistent with the majority of previous reports (22, 23). While, other studies from Nepal and Pakistan reported *C. krusei* as the most common NAC species (23). Additionally, the prevalence of *C. albicans* declined yearly with a year-on-year increase in NAC species, especially for *C. glabrata*. This trend signals an important epidemiological shift and is consistent with recent epidemiological studies (17, 22). Existing evidence suggests that the excessive use of antifungal agents may contribute to the selection of antifungal-resistent NAC species (24).

We also observed that the percentage of VVC was highest among women aged 19–35 years (65.09%), similar to Nepal (25), India (26), and Saudi Arabia (27). This high incidence in the reproductive age group may be associated with higher levels of sexual activity (25). Additionally, we found that women at or near menopause face a high risk of NAC infection compared to younger individuals. Whether this association is driven by hormonal changes, altered immune responses, or some other factor warrants further investigation (28).

Research has demonstrated that AMB exhibits potent activity against complicated VVC (29). In the present study, 99.97% of isolates were susceptible to AMB, aligning with existing literature (28, 30). Similarly, the rate of resistance to 5-fluorocytosine among *Candida* isolates was very low, with 95.92% exhibiting susceptibility. These findings underscore the efficacy of both amphotericin B and 5-fluorocytosine against *Candida* infections, including NAC species. However, variable resistance to azole antifungals was observed among the *Candida* spp. isolates. Overall, fluconazole resistance was observed in 16.35% of isolates, and cross-resistance within the azole class was also detected. Specifically, of the 7,894 *Candida* isolates analyzed, 3,043 (38.55%) exhibited resistance/NWT phenotypes to at least one azole antifungal agent. Furthermore, 559 were resistant/NWT to 2 azole drugs, and 798 isolates were resistant/NWT to 3. Recent US and UK studies reported fluconazole resistance (fluconazole-R) rates of 23%–30% (10,

11), a range higher than both our cohort and previous reports, likely due to greater azole exposure in these high-risk populations (28, 31, 32). Additionally, prior azole use may be linked to emerging resistance. One study found that 7% of women developed fluconazole-R *C. albicans* after more than 12 months of self-administered clotrimazole (10).

Fluconazole resistance in *C. albicans* has persisted for over a decade (17). In this study, 18.82% of isolates were fluconazole-R, significantly higher than the global range of 3.5%–8.8% (33, 34). Similarly, itraconazole resistance (41.84%) was markedly higher than the 2.4%–6.5% range previously reported in Turkey, Iran, and Vietnam (28). Regarding *C. glabrata*, our study revealed that 76.92% of fluconazole-R isolates were also cross-resistant to itraconazole and voriconazole. Additionally, the majority (98.13%) exhibited SDD to fluconazole, suggesting that *in vivo* efficacy may be suboptimal. Researches have been conducted to assess the efficacy of ATB FUNGUS 3 in comparison to the CLSI broth microdilution method. Zhang et al. evaluated the performance of the ATB FUNGUS 3 strips (both read visually and automatically) in relation to the CLSI broth microdilution method for the *in vitro* antifungal susceptibility testing of *Candida* isolates from multicenters in China. They found that the overall essential agreement between ATB visual readings and broth microdilution was 99.1%, while a high pseudo-resistant rate to azoles was observed for ATB automated readings. And lower agreement was observed for fluconazole and itraconazole, particularly with *C. tropicalis* and *C. albicans*, which was most likely due to the trailing effect of azoles (35). However, Yao et al. detected the fluconazole susceptibility of 59 clinical *C. glabrata* isolates by the broth microdilution method and the ATB FUNGUS 3 strip. They found the results were consistent (36). Figure 3 depicts the evolving antifungal profiles of VVC cases at our hospital over recent years, highlighting a concerning azole resistance. The GM MIC values of fluconazole, itraconazole, and voriconazole for *C. albicans* mostly declined from 2019 to 2023. The observed decline in resistance may be due to changes in clinicians' antimicrobial therapy regimens. Additionally, the decline in outpatient visits during the COVID-19 pandemic beginning in 2020 may have influenced the resistance data. In particular, during the Shanghai lockdown from March to May 2022, the sharp reduction in outpatient visits led many affected women to self-medicate, and the resulting increase in antifungal use likely contributed to changes in resistance patterns. However, by 2024, the GM values had returned to levels comparable to those observed when the study began (2019). This shift may reflect emerging resistance associated with altered host immunity or an increased susceptibility to fungal infection following the COVID-19 pandemic (37). And, the subsequent post-pandemic rise in outpatient visits likely fueled the increase in antifungal resistance. Ongoing monitoring of antifungal susceptibility over time (2025 and beyond) will be essential to further clarify this trend. In contrast, although *C. glabrata* showed low resistance to fluconazole (0.55%–3.27%) and itraconazole NWT rates (1.64%–8.98%), but high NWT rates to voriconazole (20.22%–27.50%), highlighting distinct species-specific resistance patterns. Other NAC species, particularly *C. krusei*, are frequently reported to exhibit elevated resistance to various antifungal agents (38). Fortunately, only two isolates (2.53%) in this study exhibited resistance to voriconazole and itraconazole. Among *C. tropicalis*, MIC values for the tested antifungals were generally higher compared to *C. albicans*. This trend may be attributed to the use of broad-spectrum antifungals, which can promote the selection of resistant *Candida* mutants.

Our study has several limitations. First, the generalizability of the findings may be constrained by the geographic concentration of sampling and the use of the ATB FUNGUS 3 method for antifungal susceptibility testing. The ATB FUNGUS 3 offers a simple, rapid, and broad-spectrum approach to antifungal susceptibility testing with good agreement for 5-fluorocytosine and amphotericin B susceptibility in *Candida* spp., while its reliability for azole susceptibility (especially in *C. albicans* and *C. glabrata*) is questionable and needs validation via the CLSI broth microdilution reference method. Second, we lacked detailed patient data, including occupation, diet, and treatment

history that could influence yeast distribution and antifungal susceptibilities. Among them, a key limitation is the absence of treatment data for these patients. Many relevant therapies are topical or combined with oral agents. While AFST breakpoints are based on systemic drug levels, topical application can achieve high local concentrations. Therefore, an organism deemed "resistant" may still be treatable with a topical antifungal. This important consideration should be noted when interpreting AFST results. Future multi-center and longitudinal studies incorporating broader geographic representation, molecular assays, and comprehensive patient data are needed to better understand VVC epidemiology and resistance dynamics.

In summary, our study underscores the dynamic epidemiology and growing challenge of antifungal resistance of VVC in Shanghai, marked by a rising prevalence of NAC species. These findings further reinforce the necessity of routine species identification and continuous antifungal susceptibility testing to optimize treatment management.

## MATERIALS AND METHODS

### Study design

This study was carried out at the Obstetrics and Gynecology Hospital of Fudan University in Shanghai from January 2019 to December 2024. Collected data included patients' age, sampling dates, and laboratory-confirmed microbiological identification and antifungal results. The study protocol was approved by the ethics committee of the Obstetrics and Gynecology Hospital of Fudan University (2024-172). The condition of harboring *Candida* species in the vagina without symptoms is defined as vaginal *Candida* colonization (VCC), while VVC is symptomatic. The entry criteria of this study included (i) the presence of pseudohyphae or budding yeast defined by wet mount microscopy, (ii) initial diagnosis of vaginitis by a gynecologist, (iii) presentation of typical clinical symptoms consistent with VVC (e.g., curd-like discharge, intense pruritus, burning, vulvar edema, dyspareunia) (39). Patients who met at least two of the above criteria were enrolled, and the vaginal swab specimen of each enrolled patient was collected and inoculated on blood agar and chromogenic agar (Comagal Microbial Technology Co., Ltd., Shanghai, China) and incubated at 28°C for 72 h. *Candida* culture positive patients were considered VVC infection and those who discovered ≥2 different *Candida* spp. isolated during the course were defined as mixed VVC infection.

### Microbial identification

After 72 h of culture, a single colony was identified by matrix-assisted laser desorption/ionization time-of-flight mass spectrometry (MALDI-TOF MS; Bruker Daltonics, Bremen, Germany). All isolates were stored at −80°C in broth with 50% glycerol.

### *In vitro* antifungal susceptibility testing

AFST of each *Candida* isolate was conducted using commercial method: ATB FUNGUS 3 strip (BioMérieux, France) following the manufacturer's instructions (40). The strip contains five antifungal agents at gradient concentrations, including 5-fluorocytosine (5-FC) (4–16 mg/L), amphotericin B (AMB) (0.5–16 mg/L), fluconazole (FLC) (1–128 mg/L), itraconazole (ITC) (0.125–4 mg/L), and voriconazole (VRC) (0.06–8 mg/L). The strip was incubated at 35 ± 2°C for 24 h, and the minimal inhibitory concentration (MIC) was determined as the lowest drug concentration that inhibits 100% (for AMB) or 50% (for the other four drugs) of fungal growth compared with the drug-free control. *Candida krusei* ATCC 6258 and *Candida parapsilosis* ATCC 22,019 were used as quality control strains.

*In vitro* susceptibility was determined according to the CLSI M27M44S-Ed3 clinical breakpoints (CBPs) for fluconazole and voriconazole (41). For amphotericin B and

itraconazole, the species-specific epidemiological cut-off values (ECVs) based on the CLSI M57S-Ed4 were applied (42). The CBPs were used to classify *Candida* isolates into susceptible (S), intermediate (I)/susceptible dose-dependent (SDD), or resistant (R), while ECVs were applied to classify isolates as either wild-type (WT; MIC ≤ ECV) or non-wild-type (NWT; MIC > ECV). Due to the lack of CLSI CBPs or ECVs for 5-FC on all *Candida* spp., the MICs of 5-FC were interpreted following the ATB FUNGUS 3 manufacturer's instructions. Given the absence of established threshold values for voriconazole on *C. lusitaniae* in current CLSI guidelines or instructions, caution is required when interpreting the susceptibility and resistance profiles. In addition, since *C. krusei* is intrinsically resistant to fluconazole, the MICs were recorded as "IR."

## Statistical analysis

Data analyses were conducted using GraphPad Prism (version 9.0, USA). The MIC, MIC range, geometric mean (GM) MIC, $MIC_{50}/MIC_{90}$ (the concentrations that inhibit 50% and 90% of isolates, respectively), and resistance/NWT rates were determined. Continuous variables were presented as mean ± standard deviation (SD) or median (interquartile range, IQR), based on their distribution, whereas categorical variables were summarized as numbers and percentages. Categorical variables were analyzed using the Chi-square test or two-tailed Fisher's exact test. The prevalence of different *Candida* species and their antifungal resistance profiles were assessed annually during the study period using the $\chi^2$ test for trend. A two-tailed *P* value of <0.05 was considered statistically significant.

## ACKNOWLEDGMENTS

This work was supported by the Clinical Research Program of the Shanghai Municipal Health Commission (202340223) and the Shanghai Natural Science Foundation Project (23ZR1408100).

Y.C. designed the study and wrote the manuscript. L.L., Q.W., and X.L. collected the data and conducted the statistical analysis. J.G. and C.Y. assisted in revising the manuscript. All the authors have read and approved the final manuscript.

## AUTHOR AFFILIATION

[1]Obstetrics and Gynecology Hospital of Fudan University, Shanghai Key Lab of Reproduction and Development, Shanghai Key Lab of Female Reproductive Endocrine Related Diseases, Shanghai, China

## AUTHOR ORCIDs

Yisheng Chen  http://orcid.org/0000-0003-2116-2458
Jing Gao  http://orcid.org/0000-0003-2711-6698
Chunmei Ying  http://orcid.org/0000-0001-7405-0834

## FUNDING

| Funder | Grant(s) | Author(s) |
| --- | --- | --- |
| Shanghai Municipal Health Commission | 202340223 | Chunmei Ying |
| Natural Science Foundation of Shanghai Municipality | 23ZR1408100 | Jing Gao |

## AUTHOR CONTRIBUTIONS

Yisheng Chen, Data curation, Formal analysis, Writing – original draft | Loukaiyi Lu, Investigation, Methodology | Qiang Wang, Investigation, Methodology | Xufan Luo, Investigation, Methodology | Jing Gao, Validation, Writing – review and editing | Chunmei Ying, Funding acquisition, Validation, Writing – review and editing

## DATA AVAILABILITY

The data sets used and analyzed during the current study are available from the corresponding author on reasonable request.

## ETHICS APPROVAL

The present study was approved by the ethics committee of the Obstetrics and Gynecology Hospital of Fudan University (2024-172) in accordance with the Declaration of Helsinki. Informed consent was obtained from all individual participants included in the study.

## ADDITIONAL FILES

The following material is available online.

### Open Peer Review

**PEER REVIEW HISTORY (review-history.pdf).** An accounting of the reviewer comments and feedback.

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
