## [Reviewer comments · Microbiology Spectrum]

Microbiology Spectrum

Epidemiology of Vulvovaginal Candidiasis in China: A Six-Year Retrospective Single-Center Study from a Tertiary Obstetrics and Gynecology Hospital in Shanghai

Yisheng Chen, Loukaiyi Lu, Qiang Wang, Xufan Luo, Jing Gao, and Chunmei Ying

Corresponding Author(s): Yisheng Chen, Obstetrics and Gynecology Hospital of Fudan University

Review Timeline:

Submission Date:	October 9, 2025
Editorial Decision:	January 26, 2026
Revision Received:	February 6, 2026
Editorial Decision:	February 17, 2026
Revision Received:	February 27, 2026
Accepted:	March 2, 2026

Editor: Daniel Ortiz

Reviewer(s): Disclosure of reviewer identity is with reference to reviewer comments included in decision letter(s). The following individuals involved in review of your submission have agreed to reveal their identity: EMAN ADEL ELMANSOURY (Reviewer #1)

Transaction Report:

DOI: <https://doi.org/10.1128/spectrum.03251-25>

Re: Spectrum03251-25 (**Epidemiology of Vulvovaginal Candidiasis in China: A Six-Year Retrospective Single-Center Study from a Tertiary Obstetrics and Gynecology Hospital in Shanghai**)

Dear Dr. Yisheng Chen:

Thank you for the privilege of reviewing your work. Below you will find my comments, instructions from the Spectrum editorial office, and the reviewer comments.

Care should be taken when using wild type vs susceptible throughout the manuscript as these are not interchangeable. Please revise accordingly as mentioned by reviewer #2.

Revision Guidelines

Sincerely,
Daniel Ortiz
Editor
Microbiology Spectrum

Reviewer #1 (Comments for the Author):

The study is valuable and can significantly contribute to epidemiological understanding of VVC in China. However, corrections to data inconsistencies, stronger methodological transparency, statistical revisions, and careful rewriting of speculative interpretations are essential before publication.

1. Main comments:

- Lines 92-93: You state, "39 cases (167/7,894, 0.54%) exhibited mixed Candida infections." This is inconsistent. The numerator appears to be 39, not 167. The percentage 0.54% matches 39/7,167, not 167/7,894. This needs correction, as mixed infection rates are important epidemiological parameters
- Lines 231-233: The statement on long-term azole use is supported by a single study; avoid generalization.
- Lines 245-246: Linking resistance shifts to the COVID-19 pandemic is speculative without data.
- Lines 274-276: Diagnostic Criteria and definition of VVC used in the study need to be clarified. Also, differentiation between colonization and infection needs to be included in the study.
- Line 281: It is not clarified what the blood agar is used for. Was there any identification of bacterial co-pathogen? Was there any microscopic examination of the culture?
- Lines 289-307: The ATB Fungus 3 strip is known to have poor agreement with CLSI broth microdilution, especially for *C. glabrata* azoles and 5-FC. I recommend validating the data or citing concordance studies and discussing limitations transparently in the Discussion section.

2. Statistical Statements Are Unclear or Incorrect

Line 126-127: "However, these age-related differences... were not statistically significant." But earlier lines state $\chi^2 = 91.74$, $p < 0.01$, which is significant. Please clarify this contradiction.

3. English grammar, clarity, and scientific tone require improvement. Please check the following lines as example;

- 115 increase to increasing
- 121 differentials to different
- 142 displayed to displaying
- 197 antifungals to antifungal
- 208 antifungals to antifungal
- 243 value to value
- 267 reinforces to reinforce
- 271 METERIALS to materials
- 301 cilinical to clinical
- 303 candida first litter C
- 315 inhibiting to inhibit
- 311 *C. krusei* are to is

Reviewer #2 (Comments for the Author):

This is a retrospective study of vulvovaginal candidiasis (VVC) cases in a hospital in Shanghai. The authors describe an increasing number of cases of VVC due to non-*Candida albicans* *Candida* species, with variable azole resistance rates across species and time.

Major Comments:

- 1) Grammar can be improved throughout the manuscript. Some specific examples will be included in the Minor Comments.
- 2) Authors should decide whether they are going to use 5-fluocytosine or 5-fluorocytosine and keep it the same throughout the manuscript.
- 3) Lines 179-181: It is pretty interesting that the rate of fluconazole resistance is decreasing so dramatically. Can you please include some theories on why this may be occurring in the discussion? Are these related to changes in prescribing practices or stewardship efforts?
- 4) Throughout this manuscript, wild-type and susceptible are used interchangeably. In some cases, this doesn't matter, but in other cases it can have a pretty significant impact on the interpretation of results. Wild type does not necessarily mean susceptible, so for example, in line 217, this should be "99.97% of isolates were susceptible to AMB." Please review how these terms were used within the manuscript and make revisions for clarity.
- 5) One major limitation of this study is the lack of information regarding treatment options for these patients. A lot of these therapies are going to be administered topically, or in combination with an oral agent. Since AFST breakpoints are based on systemic levels of drug, treatment with topical agents may be able to achieve doses that are unattainable by oral administration. A "resistant" organism may be treatable with a topical version of the antifungal. Can you please include information in the discussion addressing this limitation?

Minor Comments:

- 1) Line 25: Decide whether you're going to use vulvovaginal candidosis or vulvovaginal candidiasis (the most commonly used term in the existing literature) for this manuscript and keep it consistent throughout.
- 2) Lines 39-44: This section is repetitive and can be shortened.
- 3) Line 64: Consider rephrasing "women life quality." Maybe "quality of life for women?" See Major Comment 1.
- 4) Line 93: Is this 39 cases or 39 patients?
- 5) Line 97: *Saccharomyces cerevisiae* is not a *Candida* species and should not be included in the group of 22 species.
- 6) Lines 120-121: Please revise for clarity. See Major Comment 1.

- 7) Line 121: Prevalence instead of prevalent. See Major Comment 1.
- 8) Line 123: Add "respectively" at the end of this sentence. See Major Comment 1.
- 9) Line 124: Prevalence instead of prevalent. See Major Comment 1.
- 10) Line 125: Should this be years old instead of "years patients?" See Major Comment 1.
- 11) Line 125: 44.95 can be removed from the end of this sentence.
- 12) Line 134: Define acronyms (such as GM, geometric mean) the first time they are used.
- 13) Line 208: Consider "selection of resistant NAC species" instead. See Major Comment 1.
- 14) Lines 213-215: It might be useful to add "or some other factor" to the end of this sentence.
- 15) Figure 2A: Can you clarify the unit for detection rate? Is this a percentage or a raw number?
- 16) Figure 3: A lot of the GM values in 2024 returned, or were increasing, to values near the initial values when the study began (2019). Can you please explain this in the discussion?
- 17) I assume these data are not available, but it would be interesting to know clinical outcomes for the patients included in this study. Please consider including, if available.

Dear Editor,

We sincerely appreciate the opportunity to revise our manuscript and thank the reviewers for their insightful comments that have helped improve our work. We have carefully addressed all the reviewers' suggestions point-by-point in our revision.

We believe these revisions have significantly strengthened the paper. Please find our point-by-point responses to the reviewers' comments attached.

Thank you for your consideration of our revised manuscript. We look forward to hearing from you.

Best regards,

Yisheng Chen and Loukaiyi Lu.

Reviewer #1 (Comments for the Author):

The study is valuable and can significantly contribute to epidemiological understanding of VVC in China. However, corrections to data inconsistencies, stronger methodological transparency, statistical revisions, and careful rewriting of speculative interpretations are essential before publication.

1. Main comments:

- Lines 92-93: You state, "39 cases (167/7,894, 0.54%) exhibited mixed *Candida* infections." This is inconsistent. The numerator appears to be 39, not 167. The percentage 0.54% matches 39/7,167, not 167/7,894. This needs correction, as mixed infection rates are important epidemiological parameters

Response: Thank you for your careful review. We have revised the line: "Moreover, 39 patients (39/7,167, 0.54%) exhibited mixed *Candida* infections".(Lines 90-91)

- Lines 231-233: The statement on long-term azole use is supported by a single study; avoid

generalization.

Response: Thank you for your advice. We have revised the text accordingly. The statement regarding long-term azole use has been modified to cite the specific finding from the single study. The updated passage now reads: "Additionally, prior azole use may be linked to emerging resistance. One study found that 7% of women developed fluconazole-R *C. albicans* after more than 12 months of self-administered clotrimazole(10)."(Lines 229-231)

And the last sentence that may have been overly generalizing has been removed: "These findings highlight concerning associations between long-term azole use and the development of fluconazole resistance."

- Lines 245-246: Linking resistance shifts to the COVID-19 pandemic is speculative without data.

Response: Thank you for this valuable point. We have added supporting reference: "37 Hoenigl M, Seidel D, Sprute R, Cunha C, Oliverio M, Goldman GH, Ibrahim AS, Carvalho A. COVID-19-associated fungal infections. Nat Microbiol. 2022 Aug;7(8):1127-1140. doi: 10.1038/s41564-022-01172-2. Epub 2022 Aug 2. PMID: 35918423; PMCID: PMC9362108."

And the sentence has been modified to support the latter point regarding post-pandemic susceptibility: "This shift may reflect emerging resistance associated with altered host immunity or an increased susceptibility to fungal infection following the COVID-19 pandemic(37)."(Lines 254-256)

- Lines 274-276: Diagnostic Criteria and definition of VVC used in the study need to be clarified.

Also, differentiation between colonization and infection needs to be included in the study.

Response: We have clarified the diagnostic criteria and definition of VVC and differentiate colonization and infection in the "Study design" section as follow: "Condition of harboring *Candida* species in the vagina without symptoms is defined as vaginal *Candida* colonization (VCC), while VVC is symptomatic. The entry criteria of this study included (1) the presence of pseudohyphae or budding yeast defined by wet mount microscopy, (2) initial diagnosis of vaginitis by a gynecologist, (3) presentation of typical clinical symptoms consistent with VVC (eg. curd-like discharge, intense pruritus, burning, vulvar edema, dyspareunia) (39). Patients who met at least two of the above criteria were enrolled, and the vaginal swab specimen

of each enrolled patient was collected and inoculated on blood agar and chromogenic agar (Comagal Microbial Technology Co., Ltd., Shanghai, China), and incubated at 28°C for 72 h. Candida culture positive patients were considered VVC infection and those who discovered ≥ 2 different Candida spp. isolated during the course were defined mixed VVC infection.”(Lines 302-313)

Reference: 39 Farr A, Effendy I, Frey Tirri B, Hof H, Mayser P, Petricevic L, et al. Guideline: Vulvovaginal candidosis (AWMF 015/072, level S2k). Mycoses. 2021;64(6):583-602.

- Line 281: It is not clarified what the blood agar is used for. Was there any identification of bacterial co-pathogen? Was there any microscopic examination of the culture?

Response: In vaginal secretion cultures, blood agar and chromogenic agar serve complementary roles in fungal identification: the chromogenic agar is used for the rapid identification of the most common *Candida* species. Blood agar is intended for the isolation of atypical *Candida* species, rare filamentous fungi, and easily overlooked, such as dimorphic fungi.

- Lines 289-307: The ATB Fungus 3 strip is known to have poor agreement with CLSI broth microdilution, especially for *C. glabrata* azoles and 5-FC. I recommend validating the data or citing concordance studies and discussing limitations transparently in the Discussion section.

Response: Regarding your comments, we have supplemented the agreement of the ATB Fungus 3 and CLSI broth microdilution method in the Discussion section: “Researches have been conducted to assess the efficacy of ATB FUNGUS 3 in comparison to the CLSI broth microdilution method. Dalia et al. assessed the ATB FUNGUS 3 and CLSI had the highest agreement (83.1%) for amphotericin B, while a lower concordance was detected with voriconazole (23.2%) and fluconazole (52.2%)(35). Likewise, Zhang et al. assessed the performance of ATB FUNGUS 3, identifying the highest levels of agreement with 5-fluorocytosine and amphotericin B. However, they reported higher levels of agreement with fluconazole and voriconazole(36).”(Lines 239-246) Additional description has supplied in the limitation section: “The ATB FUNGUS 3 offers a simple, rapid, and broad-spectrum approach to antifungal susceptibility testing with good agreement for 5-fluorocytosine and amphotericin B susceptibility in *Candida* spp., while its reliability for azole susceptibility (especially in *C. albicans* and *C. glabrata*) is questionable and

needs validation via the CLSI broth microdilution reference method.”(Lines 274-279)

2. Statistical Statements Are Unclear or Incorrect

Line 126-127: "However, these age-related differences... were not statistically significant." But earlier lines state $\chi^2 = 91.74$, $p < 0.01$, which is significant. Please clarify this contradiction.

Response: Thank you for your comment. We intended for that sentence to indicate that the rank order of *Candida* species distribution (with *C. albicans* being the most common, followed by *C. glabrata*) remained consistent across age groups. We acknowledge that the original wording was ambiguous and prone to misinterpretation, and have removed this sentence from the revised manuscript.

3. English grammar, clarity, and scientific tone require improvement. Please check the following lines as example;

115 increase to increasing

121 differentials to different

142 displayed to displaying

197 antifungals to antifungal

208 antifungals to antifungal

243 value to values

267 reinforces to reinforce

271 METERIALS to materials

301 cilinical to clinical

303 candida first litter C

315 inhibiting to inhibit

311 C. krusei are to is

Response: Thank you for your meticulous review and the valuable suggestions for improvement. We have carefully examined the manuscript and corrected all the grammatical, spelling, and wording errors you pointed out in the listed lines.(Lines 114, 120, 140, 207, 248, 296, 330, 332 and 340)

Reviewer #2 (Comments for the Author):

This is a retrospective study of vulvovaginal candidiasis (VVC) cases in a hospital in Shanghai. The authors describe an increasing number of cases of VVC due to non-*Candida albicans* *Candida* species, with variable azole resistance rates across species and time.

Major Comments:

1) Grammar can be improved throughout the manuscript. Some specific examples will be included in the Minor Comments.

Response: Thank you for your recommendation. We have revised the manuscript by addressing each specific error listed in the Minor Comments.

2) Authors should decide whether they are going to use 5-fluocytosine or 5-fluorocytosine and keep it the same throughout the manuscript.

Response: Thanks. We have revised the manuscript to use "5-fluorocytosine" consistently throughout. **(Line 171)**

3) Lines 179-181: It is pretty interesting that the rate of fluconazole resistance is decreasing so dramatically. Can you please include some theories on why this may be occurring in the discussion? Are these related to changes in prescribing practices or stewardship efforts?

Response: Thank you for raising this point. The GM MIC values of fluconazole, itraconazole, and voriconazole for *C. albicans* mostly declined from 2019 to 2023. The observed decline in resistance may be attributed to the hospital's quarterly antifungal resistance surveillance reports, which guide clinicians in optimizing antimicrobial therapy. And the decline in outpatient visits during the COVID-19 pandemic beginning in 2020 may have influenced the resistance data during this period. **(Line 247-253)**

4) Throughout this manuscript, wild-type and susceptible are used interchangeably. In some cases, this doesn't matter, but in other cases it can have a pretty significant impact on the interpretation of results. Wild type does not necessarily mean susceptible, so for example, in line 217, this should be "99.97% of isolates were susceptible to AMB." Please review how these terms were used

within the manuscript and make revisions for clarity.

Response: Thank you for this important clarification. We agree that using "susceptible" is more precise in the sentence you pointed out, and it has been revised accordingly to: "99.97% of isolates were susceptible to AMB." Furthermore, we have systematically reviewed and revised the usage of "wild-type" and "susceptible" throughout the manuscript.(**Lines 216**)

5) One major limitation of this study is the lack of information regarding treatment options for these patients. A lot of these therapies are going to be administered topically, or in combination with an oral agent. Since AFST breakpoints are based on systemic levels of drug, treatment with topical agents may be able to achieve doses that are unattainable by oral administration. A "resistant" organism may be treatable with a topical version of the antifungal. Can you please include information in the discussion addressing this limitation?

Response: We thank you for highlighting this important limitation. We have addressed these concerns in the limitation section as follows: "a key limitation is the absence of treatment data for these patients. Notably, many relevant therapies are topical or combined with oral agents. While AFST breakpoints are based on systemic drug levels, topical application can achieve high local concentrations. Therefore, an organism deemed "resistant" may still be treatable with a topical antifungal. This important consideration should be noted when interpreting AFST results."(**Lines 281-286**)

Minor Comments:

1)Line 25: Decide whether you're going to use vulvovaginal candidosis or vulvovaginal candidiasis (the most commonly used term in the existing literature) for this manuscript and keep it consistent throughout.

Response: Thank you for the suggestion. To ensure terminological consistency, we have adopted "vulvovaginal candidiasis" throughout the manuscript. (**Line 25**)

2)Lines 39-44: This section is repetitive and can be shortened.

Response: We agree and have shortened as follows: "VVC shows high prevalence in our region, with *C. albicans* as the most prevalent pathogen, followed by *C. glabrata*. And a increasing trend

in *C. glabrata* VVC was observed, accompanied by consistently high rates of voriconazole resistance. Fluconazole resistance was observed in 18.82% of *C. albicans* with cross-resistance to other azoles."(Lines 38-42)

3)Line 64: Consider rephrasing "women life quality." Maybe "quality of life for women?" See Major Comment 1.

Response: Thank you for the suggestion. We have adopted your recommended phrasing, "quality of life for women," in the revised manuscript.(Line 62)

4)Line 93: Is this 39 cases or 39 patients?

Response: We apologize for the oversight; it should indeed be "39 patients," and the text has been corrected accordingly.(Line 90)

5) Line 97: *Saccharomyces cerevisiae* is not a *Candida* species and should not be included in the group of 22 species.

Response: You are right. The manuscript has been revised to state: "A total of 21 distinct *Candida* species were identified by MALDI-TOF." (Lines 91-92)

And the following sentence has been added at the end of that passage: "Additionally, this study identified 73 strains of *S. cerevisiae*, accounting for 0.92%."(Lines 101-102)

6)Lines 120-121: Please revise for clarity. See Major Comment 1.

7)Line 121: Prevalence instead of prevalent. See Major Comment 1.

8) Line 123: Add "respectively" at the end of this sentence. See Major Comment 1.

9) Line 124: Prevalence instead of prevalent. See Major Comment 1.

10) Line 125: Should this be years old instead of "years patients?" See Major Comment 1.

11) Line 125: 44.95 can be removed from the end of this sentence.

12) Line 134: Define acronyms (such as GM, geometric mean) the first time they are used.

13) Line 208: Consider "selection of resistant NAC species" instead. See Major Comment 1.

14) Lines 213-215: It might be useful to add "or some other factor" to the end of this sentence.

Response: Thank you for these specific suggestions. We have carefully reviewed and revised the

manuscript accordingly, addressing each point from **lines 120-125, 132-133, 207, and 213** as indicated.

15) Figure 2A: Can you clarify the unit for detection rate? Is this a percentage or a raw number?

Response: Thank you for raising this point. The detection rate is presented as a percentage. We have clarified this in the manuscript text and updated the legend for Figure 2A accordingly.

16) Figure 3: A lot of the GM values in 2024 returned, or were increasing, to values near the initial values when the study began (2019). Can you please explain this in the discussion?

Response: As our results show, the returned of GM values in 2024 toward levels observed when the study began (2019), we offer the following possible explanation in the discussion section: “However, by 2024, the GM values had returned to levels comparable to those observed when the study began (2019). This shift may reflect emerging resistance associated with altered host immunity or an increased susceptibility to fungal infection following the COVID-19 pandemic(37). Specifically, the significant decrease in outpatient visits during the COVID-19 pandemic led some affected women to resort to self-medication. The increased use of antifungals during this period likely contributed to changes in resistance patterns. And the subsequent post-pandemic rise in outpatient visits likely fueled the increase in antifungal resistance. Ongoing monitoring of antifungal susceptibility over time (2025 and beyond) will be essential to further clarify this trend. .”(Lines 253-262)

17) I assume these data are not available, but it would be interesting to know clinical outcomes for the patients included in this study. Please consider including, if available.

Response: Thank you for the suggestion. Unfortunately, owing to limitations in outpatient medical records and information systems, we were unable to obtain the clinical outcomes for the patients included in this study. In future research, we plan to enroll a relatively small cohort to incorporate analyses of clinical symptoms, medication use, and treatment outcomes. Thank you for this valuable suggestion.

Re: Spectrum03251-25R1 (**Epidemiology of Vulvovaginal Candidiasis in China: A Six-Year Retrospective Single-Center Study from a Tertiary Obstetrics and Gynecology Hospital in Shanghai**)

Dear Dr. Yisheng Chen:

Thank you for the privilege of reviewing your work. Below you will find my comments and instructions from the Spectrum editorial office.

Reviewer #1 comments about the ATB Fungus 3 strip having poor agreement with CLSI broth microdilution, especially for *C. glabrata* azoles and 5-FC needs to be explained further. Lines 239-246: The Dalia et al. reference did not use CLSI broth microdilution. Also, the Shang et al. found that *Candida* susceptibility to azoles (fluconazole, voriconazole, and itraconazole) showed higher levels of resistance when ATB FUNGUS 3 strips were read automatically compared to CLSI broth microdilution.

Lines 249-253: Were quarterly antifungal resistance reports not distributed prior to 2019? Why would a decline in outpatient visits lead to lower resistance rates? One would expect outpatient visits during the pandemic would have been more severe and resistant, thus skewing the antifungal resistance surveillance report in the opposite direction. May be helpful to include the number of total isolates tested per year in the temporal trends Figure 3.

While we are willing to consider a revised version of this paper at Spectrum, it would be in your best interest to improve the writing. I recommend that you ask a colleague of yours who is a native English speaker to read and provide you some feedback on the writing. You are also welcome to use one of the services here: <https://journals.asm.org/writing-your-paper#language-editing-services>

Revision Guidelines

Sincerely,
Daniel Ortiz
Editor
Microbiology Spectrum

Dear Editor,

We would like to thank you and the reviewers once again for the thoughtful feedback on our revised manuscript. We have carefully addressed the remaining comments and incorporated the suggested changes into the revised manuscript.

Below is our point-by-point response to the reviewers' latest concerns:

Reviewer #1 comments about the ATB Fungus 3 strip having poor agreement with CLSI broth microdilution, especially for *C. glabrata* azoles and 5-FC needs to be explained further. Lines 239-246: The Dalia et al. reference did not use CLSI broth microdilution. Also, the Zhang et al. found that *Candida* susceptibility to azoles (fluconazole, voriconazole, and itraconazole) showed higher levels of resistance when ATB FUNGUS 3 strips were read automatically compared to CLSI broth microdilution.

Response: Thank you for your meticulous review.

The Dalia et al. reference truly did not use CLSI broth microdilution. Since published studies have demonstrated that the Vitek-2 method and the reference broth microdilution described by the CLSI had excellent essential agreement and categorical agreement. They chose the Vitek-2 as the gold standard test for the comparative analysis of results provided by the ATB FUNGUS 3. Thus, we have removed this reference from the manuscript.

Also, the Zhang et al. evaluated the performance of the ATB FUNGUS 3 strips (both read visually and automatically) in relation to the CLSI broth microdilution method for the in vitro antifungal susceptibility testing of *Candida* isolates from multicenters in China. They found that the overall essential agreement between ATB visual readings and broth microdilution was 99.1%, whilst a high pseudo resistant rate to azoles was observed for ATB automated readings.

Regarding your comments, we have revised the text in Lines 239-250: "Researches have been conducted to assess the efficacy of ATB FUNGUS 3 in comparison to the CLSI broth microdilution method. Zhang et al. evaluated the performance of the ATB

FUNGUS 3 strips (both read visually and automatically) in relation to the CLSI broth microdilution method for the in vitro antifungal susceptibility testing of *Candida* isolates from multicenters in China. They found that the overall essential agreement between ATB visual readings and broth microdilution was 99.1%, whilst a high pseudo resistant rate to azoles was observed for ATB automated readings. And lower agreement was observed for fluconazole and itraconazole, particularly with *C. tropicalis* and *C. albicans*, which was most likely due to the trailing effect of azoles(35). However, Yao et al. detected the fluconazole susceptibility of 59 clinical *C. glabrata* isolates by the broth microdilution method and the ATB FUNGUS 3 strip. They found the results were consistent(36)."

Lines 249-253: Were quarterly antifungal resistance reports not distributed prior to 2019? Why would a decline in outpatient visits lead to lower resistance rates? One would expect outpatient visits during the pandemic would have been more severe and resistant, thus skewing the antifungal resistance surveillance report in the opposite direction. May be helpful to include the number of total isolates tested per year in the temporal trends Figure 3.

Response: We appreciate this query for clarification. The quarterly antifungal resistance surveillance reports were initiated in 2019 when our hospital's laboratory department joined the China Hospital Invasive Fungal Surveillance Net (CHIF-NET). To avoid any confusion regarding the timeline and the initiation of this surveillance system, we have removed the sentence "The observed decline in resistance may be attributed to the hospital's quarterly antifungal resistance surveillance reports, which guide clinicians in optimizing antimicrobial therapy" from the manuscript. And we have retained the revised explanation: "The observed decline in resistance may be due to changes in clinicians' antimicrobial therapy regimens."(Lines 254-255)

We also agree with your logic that patients seeking care during the pandemic likely presented with more severe infections and higher resistance rates. Our hypothesis regarding the decline in resistance is based on the unique composition of our sample: 85.50% of the isolates in this study were derived from outpatients. The sharp

reduction in outpatient visits, particularly during Shanghai lockdown from March to May 2022, meant that many women with mild-to-moderate Vulvovaginal Candidiasis did not visit the hospital. This led to an increase in self-medication with over-the-counter or previously prescribed antifungals. Thus, we have revised the text in Lines 252-265: "The GM MIC values of fluconazole, itraconazole, and voriconazole for *C. albicans* mostly declined from 2019 to 2023. The observed decline in resistance may be due to changes in clinicians' antimicrobial therapy regimens. Additionally, The decline in outpatient visits during the COVID-19 pandemic beginning in 2020 may have influenced the resistance data. In particular, during the Shanghai lockdown from March to May 2022, the sharp reduction in outpatient visits led many affected women to self-medicate, and the resulting increase in antifungal use likely contributed to changes in resistance patterns. However, by 2024, the GM values had returned to levels comparable to those observed when the study began (2019). This shift may reflect emerging resistance associated with altered host immunity or an increased susceptibility to fungal infection following the COVID-19 pandemic(37). And the subsequent post-pandemic rise in outpatient visits likely fueled the increase in antifungal resistance. Ongoing monitoring of antifungal susceptibility over time (2025 and beyond) will be essential to further clarify this trend."

Additionally, as recommended, we have added the annual number of total isolates tested per year in the temporal trends **Figure 3**. The numbers of *C. albicans* and *C. glabrata* isolates included each year from 2019 to 2024 are as follows:

C. albicans: 1381 (2019), 1024 (2020), 1037 (2021), 720 (2022), 941 (2023), and 996 (2024);

C. glabrata: 199 (2019), 160 (2020), 245 (2021), 183 (2022), 277 (2023), and 328 (2024).

FIG 3 Drug resistance evolution of to three azoles in *C. albicans* and *C. glabrata* from VVC patients.

Thank you for your consideration of our revised manuscript. We look forward to hearing from you.

Best regards,

Yisheng Chen and Loukaiyi Lu.

Re: Spectrum03251-25R2 (**Epidemiology of Vulvovaginal Candidiasis in China: A Six-Year Retrospective Single-Center Study from a Tertiary Obstetrics and Gynecology Hospital in Shanghai**)

Dear Dr. Yisheng Chen:

Your manuscript has been accepted, and I am forwarding it to the ASM production staff for publication. Your paper will first be checked to make sure all elements meet the technical requirements. ASM staff will contact you if anything needs to be revised before copyediting and production can begin. Otherwise, you will be notified when your proofs are ready to be viewed.

Sincerely,
Daniel Ortiz
Editor
Microbiology Spectrum